# The five homologous CiaR-controlled Ccn sRNAs of *Streptococcus pneumoniae* modulate Zn-resistance

**Nicholas R. De Lay**[1,2]*, **Nidhi Verma**[1], **Dhriti Sinha**[1], **Abigail Garrett**[3], **Maximillian K. Osterberg**[4], **Daisy Porter**[1], **Spencer Reiling**[1], **David P. Giedroc**[4], **Malcolm E. Winkler**[3]

**1** Department of Microbiology and Molecular Genetics, McGovern Medical School, University of Texas Health Science Center, Houston, Texas, United States of America, **2** MD Anderson Cancer Center UTHealth Graduate School of Biomedical Sciences, University of Texas Health Science Center, Houston, Texas, United States of America, **3** Department of Biology, Indiana University Bloomington, Bloomington, Indiana, United States of America, **4** Department of Chemistry, Indiana University, Bloomington, Bloomington, Indiana, United States of America

* nicholas.r.delay@uth.tmc.edu

**Data Availability Statement:** Primary data from the mRNA-seq analyses were submitted to the NCBI Gene Expression Omnibus (GEO) and have the accession number GSE246655.

## Abstract

Zinc is a vital transition metal for all bacteria; however, elevated intracellular free Zn levels can result in mis-metalation of Mn-dependent enzymes. For Mn-centric bacteria such as *Streptococcus pneumoniae* that primarily use Mn instead of Fe as an enzyme cofactor, Zn is particularly toxic at high concentrations. Here, we report our identification and characterization of the function of the five homologous, CiaRH-regulated Ccn sRNAs in controlling *S. pneumoniae* virulence and metal homeostasis. We show that deletion of all five *ccn* genes (*ccnA*, *ccnB*, *ccnC*, *ccnD*, and *ccnE*) from *S. pneumoniae* strains D39 (serotype 2) and TIGR4 (serotype 4) causes Zn hypersensitivity and an attenuation of virulence in a murine invasive pneumonia model. We provide evidence that bioavailable Zn disproportionately increases in *S. pneumoniae* strains lacking the five *ccn* genes. Consistent with a response to Zn intoxication or relatively high intracellular free Zn levels, expression of genes encoding the CzcD Zn exporter and the Mn-independent ribonucleotide reductase, NrdD-NrdG, were increased in the Δ*ccnABCDE* mutant relative to its isogenic *ccn*⁺ parent strain. The growth inhibition by Zn that occurs as the result of loss of the *ccn* genes is rescued by supplementation with Mn or Oxyrase, a reagent that removes dissolved oxygen. Lastly, we found that the Zn-dependent growth inhibition of the Δ*ccnABCDE* strain was not altered by deletion of *sodA*, whereas the *ccn*⁺ Δ*sodA* strain phenocopied the Δ*ccnABCDE* strain. Overall, our results indicate that the Ccn sRNAs have a crucial role in preventing Zn intoxication in *S. pneumoniae*.

## Author summary

Zn and Mn are essential micronutrients for many bacteria, including *Streptococcus pneumoniae*. While Zn performs vital structural or catalytic roles in certain proteins, in excess, Zn can inhibit Mn uptake by *S. pneumoniae* and displace, but not functionally replace Mn

**Funding:** This work was supported by NIGMS grant T32GM131994 (to M.K.O.), NIGMS grant R35GM118157 (to D.P.G.), NIGMS grant R35GM131767 (to M.E.W.), McGovern Medical Startup funds, and NIAID grant R21AI171771 (to N.R.D.L.). The funders had no role in study design, data collection and analysis, decision to publish, or preparation of the manuscript.

**Competing interests:** The authors have declared that no competing interests exist.

from key enzymes including superoxide dismutase A (SodA). Here, we show that the Ccn small regulatory RNAs promote *S. pneumoniae* resistance to Zn intoxication. Furthermore, we demonstrate that these small regulatory RNAs modulate the ability of *S. pneumoniae* to cause invasive pneumonia. Altogether, these findings reveal a new layer of regulation of *S. pneumoniae* Zn homeostasis and suggest that there are factors in addition to known transporters that modulate intracellular, bioavailable Zn levels.

## Introduction

Small regulatory RNAs have been established as fundamental regulators of gene expression in bacteria and are involved in controlling nearly every aspect of bacterial physiology, metabolism, and behavior [1–3]. Two basic classes of small regulatory RNAs have been identified and characterized, those that control gene expression by directly interacting with transcripts via hydrogen bonding between complementary or wobble base-pairs and others that indirectly affect transcript abundance by titrating an RNA or DNA-binding protein [4,5]. Interactions between the former class of riboregulators, henceforth referred to as sRNAs, and their cognate target transcripts can result in changes in mRNA transcription, translation, and/or stability depending on many factors including the sequence, accessibility, structure, and location of the sRNA binding site. One of the most facile yet prevalent modes of regulation involves the sRNA binding within or adjacent to the translation initiation region blocking the 16S rRNA within the 30S ribosomal subunit from base-pairing with the complementary Shine-Delgarno sequence, or ribosome binding site, within the mRNA. Many other elegant mechanisms of sRNA-based gene regulation have been uncovered [6–8]. While a large amount of progress has been made towards understanding the contribution of sRNAs to the response of Gram-negative bacteria such as *Escherichia coli* to internally and externally derived stresses, environmental cues, and host interactions, much less headway has been achieved in understanding the functions of sRNAs in Gram-positive bacteria, particularly, *Streptococcus pneumoniae* (pneumococcus).

The Gram-positive, ovoid diplococcus *S. pneumoniae* is a leading cause of lower respiratory infection morbidity and mortality worldwide resulting in nearly 2 million deaths per year [9]. We and others have discovered 100s of putative sRNAs in *S. pneumoniae* [10–15], but the functions of almost all of them remains a mystery. Among the first sRNAs identified in *S. pneumoniae* were the five homologous Ccn sRNAs (CcnA, CcnB, CcnC, CcnD, and CcnE) [15,16], which are highly similar in sequence and predicted structure; however, CcnE contains a small insertion in its 5' end. Each Ccn sRNA is primarily transcribed from its own promoter, which is activated by the CiaRH two-component system; expression of the CiaRH two-component systems is induced by penicillin and sialic acid [17,18]. Regardless, considerable variation exists in the level of transcription of each Ccn sRNA, with CcnC being transcribed at approximate 3- to 5-times higher levels than other Ccn sRNAs under some conditions [16]. Shortly after the discovery of the five Ccn sRNAs, Tsui, Mukerjee (Sinha), et al demonstrated that CcnA negatively regulates competence and the *comCDE* mRNA encoding the precursor of the competence stimulating peptide and the two-component system that responds to this signal and activates competence [15]. Schnorpfeil et al formally demonstrated that the five Ccn sRNAs negatively regulate competence by base-pairing with the *comCDE* mRNA [19]. Other likely targets post-transcriptionally regulated by the Ccn sRNAs were identified in that study including mRNAs encoding components of a galactose transporter (*spd_0090*), a formate-nitrate transporter (*nirC*), branched-chain amino acid transporter (*brnQ*) and a toxin (*shetA*), but direct regulation of these targets by the Ccn sRNAs has not yet been established [19]. One

of these five homologous sRNAs, CcnE, has also been implicated in *S. pneumoniae* strain TIGR4 virulence in a murine invasive pneumonia model [12].

Here, we report our discovery of a role for the five Ccn sRNAs in controlling *S. pneumoniae* virulence and Zn resistance. Specifically, we show that deletion of the genes encoding the five Ccn sRNAs attenuates the virulence of *S. pneumoniae* strains D39 and TIGR4 in a murine invasive pneumonia model. Additionally, we show that loss of the Ccn sRNAs leads *S. pneumoniae* D39 and TIGR4 to become hypersensitive to Zn toxicity, and this Zn hypersensitivity is alleviated by supplementation with Mn or Oxyrase, which reduces dissolved oxygen. Altogether, our results indicate that the Ccn sRNAs prevents *S. pneumoniae* Zn intoxication by reducing the intracellular abundance of free Zn, which in turn increases its resistance to oxidative stress under aerobic growth conditions as the result of an increase in the amount of active superoxide dismutase A (SodA).

## Results

### The Ccn sRNAs are important for *S. pneumoniae* pathogenesis

Work from a prior study [12] indicated that deletion of one of the five Ccn sRNA genes (*ccnE*) reduced *S. pneumoniae* serotype 4 strain TIGR4 virulence in a murine invasive pneumonia model. In that study, the authors also discovered by Tn-seq that transposon insertions in *ccnE* reduced *S. pneumoniae* strain TIGR4 fitness in murine lungs, whereas transposon insertions in *ccnA* had no significant impact on its fitness in the murine lung, nasopharynx, or blood. To determine the contribution of the Ccn sRNAs to *S. pneumoniae* virulence, we initially made single deletions of *ccnA*, *ccnB*, *ccnC*, *ccnD*, or *ccnE* and a quintuple deletion of all five *ccn* genes in the archetypal serotype 2 *S. pneumoniae* strain D39, which causes rapid killing of mice by sepsis [20]. We then determined the consequence of these deletions on *S. pneumoniae* pathogenicity in a murine invasive pneumonia model (see Materials and Methods). While removal of any single *ccn* gene had no significant impact on its virulence in mice (S1 Fig), deletion of all five *ccn* genes attenuated *S. pneumoniae* strain D39 pathogenicity increasing median survival time from 43 h to 67 h (Fig 1A). Mice that ultimately succumbed to pneumococcal

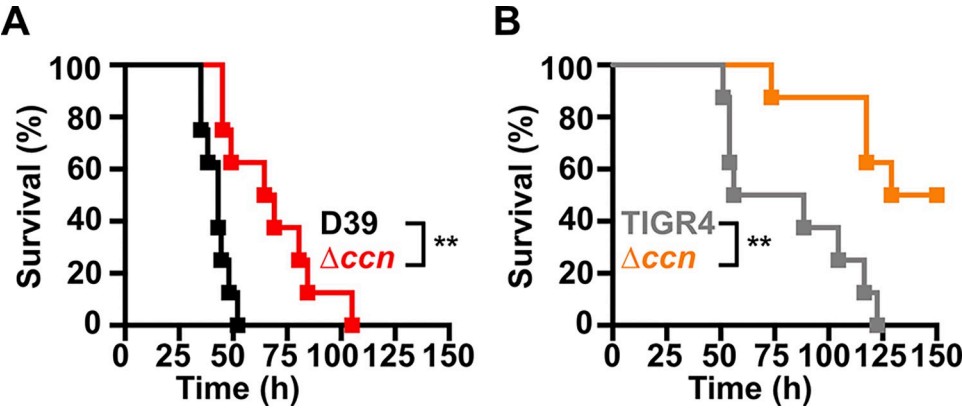

**Fig 1. Virulence phenotypes of *S. pneumoniae* strains harboring deletion of the *ccn* genes.** Survival curve of ICR outbred mice after infection with ~$10^7$ CFU in a 50 μL inoculum of the following *S. pneumoniae* strains: (A) IU1781 (D39 *rpsL1*) and NRD10176 (Δ*ccn rpsL1*); (B) NRD10220 (TIGR4 *rpsL1*) and NRD10266 (Δ*ccn rpsL1*). The difference in median survival time of IU1781 vs NRD10176 (43.0 h vs 66.8 h) and NRD10220 vs NRD10266 (72.3 h vs 139.5 h) were statistically significant. Eight mice were infected per strain. Disease progression of animals was monitored, the time at which animals reached a moribund state was recorded, and these mice were subsequently euthanized as described in Materials and Methods. A survival curve was generated from this data and analyzed by Kaplan-Meier statistics and log rank test to determine P-values, which are indicated as ** ($P < 0.005$).

infection had $\sim 10^{10}$ colony forming units of *S. pneumoniae* per mL of blood regardless of whether any of the *ccn* genes were deleted.

To confirm that the *ccn* genes are generally important for *S. pneumoniae* virulence and is not an attribute specific to strain D39, we also deleted all five *ccn* genes from the serotype 4 TIGR4 strain and measured the impact of these deletions on its virulence using the same murine invasive pneumonia model. We used strain TIGR4 for these experiments as it belongs to a different major phylogenic lineage than strain D39 [21] and has a different disease progression in mice with a propensity to cause meningitis rather than sepsis [20]. Regardless, the *ccnABCDE* deletion also resulted in a marked attenuation of *S. pneumoniae* TIGR4 virulence increasing the survival rate of ICR outbred mice from 0% to 50% (Fig 1B). While there was no significant difference in the CFUs of *S. pneumoniae* TIGR4 and the derived Δ*ccnABCDE* mutant in the blood of moribund mice, two of the mice that survived infection with the TIGR4 Δ*ccnABCDE* strain had no detectable bacteria in the blood and the other two mice had 1,000 and 2,750 CFUs per mL of blood, respectively, which was far below $\sim 10^7$ bacteria found in moribund mice that were infected with the *ccn*⁺ parent strain. Our results show that the *ccn* genes are important for *S. pneumoniae* pathogenesis.

## The Ccn sRNAs impact expression of Zn-related genes

To discover a basis for the defect in *S. pneumoniae* virulence caused by deletion of the five *ccn* genes, we compared global gene expression by high throughput RNA-sequencing (RNA-seq) between *S. pneumoniae* strain D39 or TIGR4 and the derived Δ*ccnABCDE* mutant strains grown to exponential phase ($OD_{620}$ between 0.15 and 0.2) in BHI broth at $37^{\circ}$ C in an atmosphere of 5% $CO_2$. In the *S. pneumoniae* D39 strain background, the *ccnABCDE* deletion resulted in down-regulation of 3 genes and up-regulation of 113 genes by 2-fold or more ($P_{adj} < 0.05$) (S3 Table). In contrast, deletion of the *ccn* genes from the TIGR4 strain resulted in down-regulation of 25 genes and up-regulation of 97 genes by 2-fold or greater ($P_{adj} < 0.05$) (S4 Table). 37 genes were up-regulated by 2-fold ($P_{adj} < 0.05$) in the *ccnABCDE* deletion strain in both the D39 and TIGR4 backgrounds (Table 1); among these differentially expressed genes were iron uptake system genes (*piuB*, *piuC*, *piuD*, and *piuA*), a Zn-responsive ECF (energy-coupling factor) transport gene SPD_1267/SP_1438, and *czcD* encoding a Zn/Cd exporter that provides Zn and Cd resistance. To validate our RNA-seq data, we first measured abundance of *piuB*, *spd_1267*, and *czcD* transcripts in RNA samples isolated for the RNA-seq experiment from *S. pneumoniae* strain D39 and derived Δ*ccnABCDE* strain by reverse transcriptase droplet digital PCR (RT-ddPCR). Consistent with our RNA-seq data the *piuB*, *spd_1267*, and *czcD* transcripts were up-regulated by 3.5, 10.5, and 1.9-fold respectively in the Δ*ccnABCDE* strain compared to its parental D39 strain grown in BHI broth (Fig 2A, 2B, and 2C). Using RT-ddPCR analysis of the RNA samples isolated from exponential phase cultures of *S. pneumoniae* TIGR4 and derived Δ*ccnABCDE* mutant strain grown in BHI broth at $37^{\circ}$ C under an atmosphere of 5% $CO_2$, we only observed a 1.3-fold increase in the abundance of the *czcD* mRNA in the *ccn* mutant as compared to its parental strain (Fig 2D). In *S. pneumoniae*, Zn homeostasis is intertwined with that of Mn. The ratio of Mn relative to Zn can determine whether or not a Mn- or Zn-dependent enzyme or regulator will be metalated with Mn and/or Zn, and hence be functional or inert [22–27]. In the instance of *czcD*, its transcription is activated by the transcriptional regulator SczA, when the intracellular ratio of Zn to Mn is high [24]. Thus, these RNA-seq data suggested to us that removal of the *ccn* genes from *S. pneumoniae* was leading to an increase in the intracellular free Zn concentration relative to Mn, and to cope with this stress, the *ccn* mutant strain was increasing expression of the CzcD Zn exporter.

**Table 1. Genes significantly, differentially expressed between a Δ*ccnABCDE* and *ccn*[+] strain in both the *S. pneumoniae* D39 and TIGR4 background during exponential growth in BHI broth[a].**

| D39 locus tag | Gene | Known or predicted function | D39 fold change | TIGR4 fold change |
|---|---|---|---|---|
| SPD_0025 | | tRNA-specific adenosine-34 deaminase | 84.3 | 144 |
| SPD_0027 | *dut* | deoxyuridine 5'-triphosphate nucleotidohydrolase | 3.52 | 4.39 |
| SPD_0028 | | hypothetical protein | 3.80 | 3.40 |
| SPD_0029 | *radA* | DNA repair protein | 3.55 | 2.80 |
| SPD_0090 | | galactose ABC transport protein | 2.09 | 2.00 |
| SPD_0104 | | aggregation-promoting factor | 2.69 | 2.27 |
| SPD_0222 | *gpmB1* | phosphoglycerate mutase family protein | 25.0 | 22.9 |
| SPD_0243 | *uppS* | undecaprenyl diphosphate synthase | 5.97 | 4.42 |
| SPD_0244 | *cdsA* | phosphatidate cytidylyltransferase | 5.50 | 4.49 |
| SPD_0245 | *eep* | intramembrane protease | 5.57 | 4.19 |
| SPD_0246 | *proS* | prolyl-tRNA synthetase | 5.91 | 4.68 |
| SPD_0247 | *bglA* | 6-phospho-β-glucosidase | 3.57 | 3.35 |
| SPD_0308 | *clpL* | ATP-dependent protease subunit | 13.7 | 2.71 |
| SPD_0460 | *dnaK* | protein chaperone | 3.97 | 2.01 |
| SPD_0501 | *licT* | β-glucoside operon antiterminator | 2.91 | 4.77 |
| SPD_0502 | *bglF* | β-glucoside PTS transporter subunit | 3.07 | 5.65 |
| SPD_0503 | *bglA-2* | 6-phospho-β-glucosidase | 2.57 | 4.79 |
| SPD_0615 | *glnH3* | degenerate glutamine ABC transporter subunit | 11.6 | 4.05 |
| SPD_0616 | *glnQ3* | glutamine ABC transporter subunit | 8.90 | 3.07 |
| SPD_0617 | *glnP3b* | glutamine ABC transporter subunit | 11.1 | 3.63 |
| SPD_0618 | *glnP3a* | glutamine ABC transporter subunit | 11.8 | 2.98 |
| SPD_0775 | | acetyltransferase | 3.29 | 2.71 |
| SPD_1045 | | degenerate DUF3884 domain protein | 4.73 | 3.16 |
| SPD_1046 | *lacG-2* | 6-phospho-β-galactosidase | 3.56 | 2.92 |
| SPD_1267 | | ECF transporter subunit | 11.1 | 2.14 |
| SPD_1638 | *czcD* | Cd/Zn exporter | 2.69 | 2.66 |
| SPD_1649 | *piuB* | Fe uptake transporter subunit | 5.13 | 2.43 |
| SPD_1650 | *piuC* | Fe uptake transporter subunit | 4.45 | 2.01 |
| SPD_1651 | *piuD* | Fe uptake transporter subunit | 4.22 | 2.15 |
| SPD_1652 | *piuA* | Fe uptake transporter subunit | 4.38 | 2.25 |
| SPD_1748 | *pneA2* | lantibiotic peptide | 2.19 | 2.16 |
| SPD_1749 | *lanM* | lanthionine biosynthesis protein | 2.49 | 2.29 |
| SPD_1750 | *wrbA* | FAD-dependent flavoprotein | 3.00 | 2.69 |
| SPD_1751 | | hypothetical protein | 2.56 | 3.16 |
| SPD_1752 | *clyB* | toxin secretion ABC transporter | 3.58 | 3.22 |
| SPD_1753 | | epidermin leader peptide processing serine protease | 2.44 | 2.87 |
| SPD_1932 | *malP* | malodextrin phosphorylase | 2.58 | 2.58 |

[a]RNA extraction and mRNA-seq analyses were performed as described in *Materials and Methods*. RNA was prepared from cultures of strains IU1781 (D39 *rpsL1*), NRD10176 (D39 *rpsL1* Δ*ccnABCDE*), NRD10220 (TIGR4 *rpsL1*), and NRD10266 (TIGR4 *rpsL1* Δ*ccnABCDE*) (S1 and S2 Tables). Fold changes (2.0-fold cut-off) and adjusted P-values (Pval <0.05) are based on three independent biological replicates.

## Absence of the *ccn* genes causes S. *pneumoniae* to become hypersensitive to Zn

If the absence of the *ccn* genes from *S. pneumoniae* leads to an imbalance of transition metals with higher intracellular levels of free Zn relative to Mn, then we would expect that increasing the concentration of Zn present in the medium would disproportionately impair the growth of

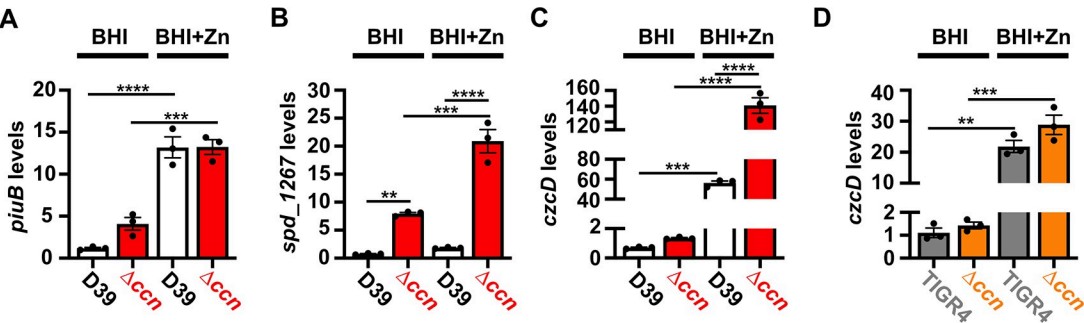

**Fig 2. Loss of the *ccn* genes perturbs the expression of Zn and Mn stress associated genes in *S. pneumoniae*.** Abundance of *piuB* (A), *spd_1267* (B), and *czcD* (C, D) mRNAs was determined by RT-ddPCR as described in Materials and Methods for strain IU1781 (D39) and derived Δ*ccnABCDE* mutant strain (NRD10176; Δ*ccn*) (A, B, and C) or strain NRD10220 (TIGR4) and derived Δ*ccnABCDE* mutant strain NRD10266 (Δ*ccn*) (D) grown to exponential phase ($OD_{620}$ of ~0.2) in BHI broth alone (BHI) or supplemented with 0.2 mM $ZnSO_4$ (BHI+Zn) at 37°C under an atmosphere of 5% $CO_2$. Transcript levels were normalized to *tuf* mRNA. Values represent the mean of three independent cultures and error bars indicate SEM. Statistical analysis was performed by ANOVA, and statistically significant results are indicated by ** ($P < 0.005$), *** ($P < 0.0005$) or **** ($P < 0.0001$).

the Δ*ccnABCDE* mutant relative to the isogenic *ccn*⁺ strain. Previous studies have indicated that Becton-Dickinson (BD) BHI broth typically contains ~20 μM Zn and 200 nM Mn [25,28]. We first compared growth of strain D39 and derived Δ*ccnA*, Δ*ccnB*, Δ*ccnC*, Δ*ccnD*, Δ*ccnE*, and Δ*ccnABCDE* strains in BHI broth alone or supplemented with 0.2 mM Zn at 37°C under an atmosphere of 5% $CO_2$. No significant difference was observed in growth rate between strain D39 and derived Δ*ccnA*, Δ*ccnB*, Δ*ccnC*, and Δ*ccnD* mutant strains in BHI in the presence or absence of 0.2 mM added Zn (S2A–S2D Fig), although the growth yield for the *ccnE* mutant was lower in BHI in the presence or absence of Zn. Growth of strain D39 and the derived Δ*ccnABCDE* mutant was similar in BHI broth alone (Fig 3A)

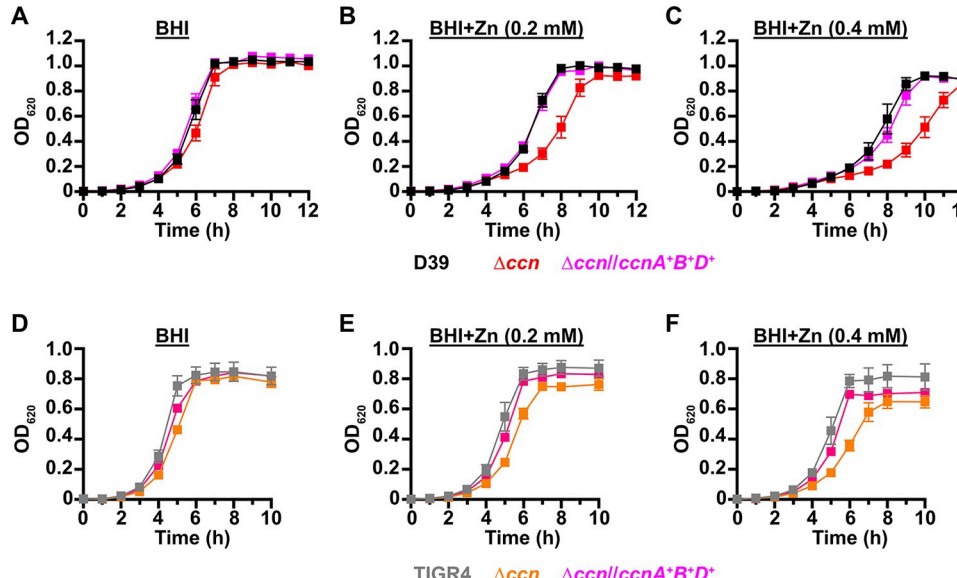

**Fig 3. Growth phenotypes of *S. pneumoniae* strains harboring deletion of the *ccn* genes.** Growth characteristics at 37°C under an atmosphere of 5% $CO_2$ in BHI broth alone (A, D) or with 0.2 mM (B, E) or 0.4 mM (C,F) $ZnSO_4$ of following strains: (A, B, C) IU1781 (D39), NRD10176 (Δ*ccn*), and NRD10396 (Δ*ccn*//*ccnA*⁺*B*⁺*D*); (D, E, F) NRD10220 (TIGR4) NRD10266 (Δ*ccn*), and NRD10787 (Δ*ccn*//*ccnA*⁺*B*⁺*D*⁺). Each point on the graph represents the mean $OD_{620}$ value from three independent cultures. Error bars, which in some cases are too small to observe in the graph, represent the standard deviation (SD).

In contrast, the absence of all five *ccn* genes led to an obvious impairment in growth rate in BHI supplemented with 0.2 mM Zn (Fig 3B). This growth deficiency relative to the *ccn*+ parental strain was also observed for the Δ*ccnABCDE* strain when Zn was increased in BHI broth to 0.4 mM (Fig 3C). Consistently, addition of Zn at 0.4 mM severely reduced the growth rate of the *ccn*+ D39 strain. We then constructed a set of strains in which every possible combination of three or four *ccn* genes are deleted and tested their growth in BHI broth alone or supplemented with 0.2 mM Zn (S2 Fig). In summary, we found that each of the strains containing only a single *ccn* gene (*ccnA*, *ccnB*, *ccnC*, *ccnD*,or *ccnE*) was defective in growth in BHI broth supplemented with Zn, but grew similar to the *ccn*+ parental strain in BHI broth alone (S2M–S2P Fig). Out of all of the strains containing only two of the five *ccn* genes, the strains expressing *ccnA* and *ccnB* (Δ*ccnCDE*) or *ccnC* and *ccnD* (Δ*ccnABE*) grew most similar to the *ccn*+ parental strain in BHI broth supplemented with Zn (S2E–S2L Fig).

Thus, we tested whether introduction of *ccnA* and *ccnB* or *ccnC* and *ccnD* expressed from their native promoters at ectopic loci restored growth of the Δ*ccnABCDE* mutant strain to that of the *ccn*+ parental strain in BHI broth with 0.2 mM added Zn, but only partial complementation was achieved (S3 Fig). Therefore, we examined whether inserting genes for three Ccn sRNAs (*ccnA*, *ccnB*, and *ccnD*) with their native promoter at ectopic loci could completely correct the Zn-dependent growth deficiency of the Δ*ccnABCDE* mutant strain, and it did (Fig 3). To verify that the Zn hypersensitivity caused by the deletion of all five *ccn* genes was not specific to the serotype 2 strain D39, we also tested the effect of the quintuple *ccn* deletion on the growth of the serotype 4 TIGR4 strain in BHI broth supplemented with Zn. Consistent with our results observed for strain D39, deletion of the *ccn* genes from TIGR4 led to growth impairment in BHI broth when Zn was added at a final concentration of 0.2 or 0.4 mM (Figs 3D–3F and S4). Moreover, the Zn dependent growth impairment of the Δ*ccnABCDE* mutant TIGR4 strain could also be fully ameliorated by ectopic expression of *ccnA*, *ccnB*, and *ccnD* (Fig 3). Curiously, Zn at the highest concentration used had less of an effect on strain TIGR4 growth than it did on strain D39. Overall, these results indicate that Ccn sRNAs promote *S. pneumoniae* Zn tolerance.

## In the absence of the Ccn sRNAs, *S. pneumoniae* accumulates bioavailable Zn

*S. pneumoniae* is a Mn-centric bacteria encoding several Mn-requiring enzymes including superoxide dismutase (SodA), a capsule regulatory kinase (CpsB), phosphoglucomutase (Pgm), phosphopentomutase (DeoB), a cell division regulating phosphatase (PhpP), an aerobic ribonucleotide reductase (NrdEF), pyruvate kinase (PyK), and lactate dehydrogenase (Ldh). Mis-metalation of these Mn-dependent enzymes by Zn, which inhibits their enzymatic activity [22,27], can occur when the internal ratio of bioavailable Zn-to-Mn is high. Additionally, the substrate binding component of the PsaBCA Mn ATP binding cassette (ABC) type transporter, the only known Mn importer in *S. pneumoniae*, has been shown to bind Zn tightly, blocking Mn uptake [26,28]. Our RNA-seq data above indicated that expression of the CzcD Zn exporter, which is expressed in response to high levels of free, or bioavailable, Zn relative to Mn [24,29], is up-regulated in *S. pneumoniae* strains lacking the *ccn* genes (Table 1 and Fig 2C and 2D). Based on these results and the published data mentioned above, we hypothesized that Zn-hypersensitivity caused by the removal of all five *ccn* genes from the *S. pneumoniae* genome is due to an increase in free Zn concentration relative to Mn. If this postulate is correct, then the Zn-dependent growth inhibition that occurs when the *S. pneumoniae* Δ*ccnABCDE* mutant strain is grown in BHI broth supplemented with 0.2 mM Zn should be rescued by inclusion of an equimolar amount of Mn into the medium. As shown in Fig 4, the growth impairment of

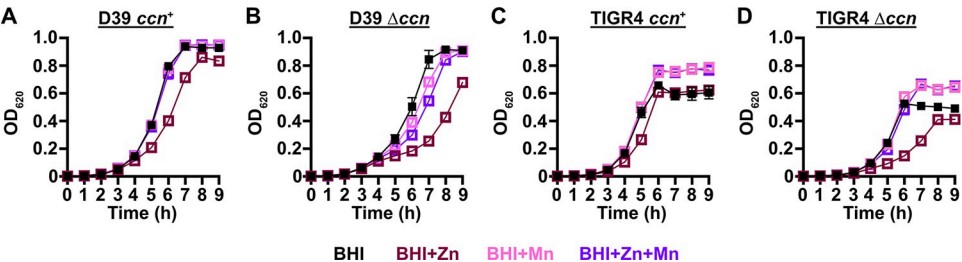

**Fig 4. Mn supplementation eliminates the Zn dependent growth inhibition of *S. pneumoniae* Δ*ccnABCDE* mutant.** Growth characteristics at 37°C under an atmosphere of 5% $CO_2$ in BHI broth alone (BHI) or with 0.2 mM $ZnSO_4$ (BHI+Zn), 0.2 mM $MnCl_2$ (BHI+Mn), or 0.2 mM $ZnSO_4$ and $MnCl_2$ (BHI+Zn+Mn) of strains (A) IU1781 (D39 *ccn*+), (B) NRD10176 (D39 Δ*ccn*), (C), NRD10220 (TIGR4 *ccn*+), and (D) NRD10266 (TIGR4 Δ*ccn*). Each point on the graph represents the mean $OD_{620}$ value from three independent cultures. Error bars, which in some cases are too small to observe in the graph, represent the standard deviation (SD).

the Δ*ccnABCDE* mutant of *S. pneumoniae* D39 or TIGR4 strain in BHI broth with 0.2 mM Zn is cured by addition of 0.2 mM Mn consistent with our model.

To directly test whether or not the levels of transition metals are perturbed in strains lacking the *ccn* genes, we measured total cell-associated transition metals in *S. pneumoniae* strain D39, derived Δ*ccnABCDE* mutant, and the Δ*ccnABCDE* strain complemented with *ccnA*, *ccnB*, and *ccnD* grown in BHI broth or the chemically defined C medium by inductively coupled plasma-mass spectrometry (ICP-MS). During exponential growth ($OD_{620}$ of ~0.2) in BHI broth alone or supplemented with 0.2 mM Zn, there was no significant difference in total cell-associated Zn among these strains (Fig 5A and 5B and Table 2). However, it remains possible that there was a difference in the amount of bioavailable, or unbound, Zn as our ICP-MS based approach measures the total amount of cell associated metals and does not discriminate between protein-bound vs unbound metals.

Since we were unable to detect a difference in Zn or Mn content among the *S. pneumoniae* strain D39 strain, Δ*ccnABCDE* mutant, and derived strain complemented with *ccnA*, *ccnB*, and *ccnD* in BHI broth, we then assessed their abundance when these strains were grown in a defined liquid medium (C-medium). Similar to what was observed in BHI broth supplemented with Zn, we found that Δ*ccnABCDE* mutant had a slower growth rate, or longer doubling time, than its parental *ccn*+ *S. pneumoniae* D39 strain in C-medium supplemented with 0.2 mM $ZnSO_4$ (65 min vs 56 min), but not in C-medium alone (47 min vs 42 min) as shown in S5 Fig. Next, we measured total cell-associated Zn and Mn of the aforementioned strains under these growth conditions, and we observed a statistically significant difference ($P < 0.05$) in the median Zn abundance between the *S. pneumoniae* strain D39 and derived Δ*ccnABCDE* mutant grown in C-medium alone (183% increase) or supplemented with Zn (144% increase) (Fig 5C and 5D and Table 2). Complementation of the Δ*ccnABCDE* mutant with *ccnA*, *ccnB*, and *ccnD* did not restore Zn levels to that of its parental strain signaling that all five *ccn* genes may be needed to maintain proper Zn homeostasis. No statistically significant difference in total cell-associated Mn was observed between *S. pneumoniae* strain D39 and derived Δ*ccnABCDE* mutant under any of the tested growth conditions (Fig 5B and 5D and Table 2). Thus, our evidence that Mn supplementation eliminated the growth deficiency of the *ccn*- strain caused by excess Zn (Fig 4), that there was increased expression of *czcD* encoding a Zn exporter when the *ccn* genes were removed from *S. pneumoniae* strains D39 and TIGR4 (Tables 1, S3, and S4), and that the amount of Zn associated with the Δ*ccnABCDE* mutant strain was higher compared to the *ccn*+ strain in C-medium (Fig 5 and Table 2) suggest that the Ccn sRNAs are important for preserving Zn homeostasis in *S. pneumoniae*.

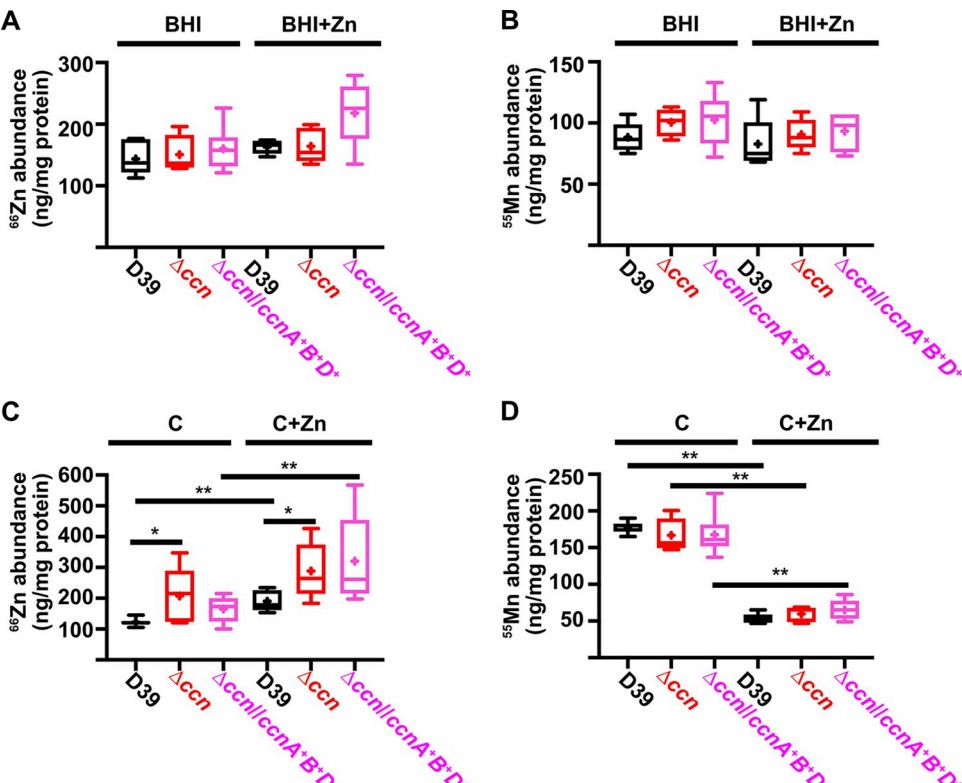

**Fig 5. Deletion of the *ccn* genes increases total cell-associated Zn levels, but not Mn levels, in C medium.** Total cell associated Zn (A, C) and Mn (B, D) abundance was measured from cells harvested from cultures of IU781 (D39), NRD10176 (Δ*ccn*), and NRD10396 (Δ*ccn*//ccnA+B+D+) grown to exponential growth phase (OD$_{620}$ of ~0.2) in BHI broth (BHI) or BHI broth with 0.2 mM ZnSO$_4$ (BHI+Zn) (A and B) or in C medium or C medium with 0.2 mM ZnSO$_4$ (C+Zn) (C and D) by ICP-MS and normalized to protein amounts. Results presented in box and whisker plots represent the median of 5 to 8 replicates with whiskers indicating the 5–95% percentile. Means are indicated by "+". Statistical analysis was performed using a Mann-Whitney test, and statistically significant results are indicated by * (P < 0.05) or ** (P <0.05).

## Oxidative stress due to reduced levels of active superoxide dismutase A contributes to the Zn hypersensitivity of the *S. pneumoniae* strain lacking the Ccn sRNAs

To discover the molecular basis for the Zn hypersensitivity caused by loss of the *ccn* genes, we turned to an RNA-seq based approach. Briefly, we compared transcript abundance in RNA

**Table 2. Total cell-associated Zn and Mn abundance of *S. pneumoniae* strain D39 and derived Δ*ccnABCDE* strains grown in BHI broth or C medium alone or with Zn supplementation[a].**

| Strain | BHI | | BHI + Zn | | C | | C + Zn | |
|---|---|---|---|---|---|---|---|---|
| | **Zn** | **Mn** | **Zn** | **Mn** | **Zn** | **Mn** | **Zn** | **Mn** |
| IU1781 (WT) | 140 ±11 (6) | 88 ±4.9 (6) | 160 ±5 (5) | 83 ±9.4 (5) | 120 ±4 (8) | 180 ±3 (8) | 190 ±15 (5) | 53 ±3 (5) |
| NRD10176 (Δ*ccn*) | 150 ±12 (6) | 100 ±4.4 (6) | 160 ±13 (5) | 91 ± 5.7 (5) | 210 ±41 (5) | 170 ±10 (5) | 290 ±41 (5) | 60 ±5 (5) |
| NRD10396 (Δ*ccn*//ccnA+B+D+) | 160 ±15 (6) | 100 ±8.6 (6) | 220 ±21 (6) | 93 ± 6.1 (6) | 170 ±17 (6) | 170 ±12 (6) | 320 ±66 (5) | 65 ±6 (5) |

[a]The indicated strains were cultured and ICP-MS analyses of metal abundance of cells harvested from these cultures was performed as described in *Materials and Methods*. Shown are the mean values of the Zn and Mn abundance normalized to protein levels (ng/mg of protein) followed by the standard error of the mean. Number of biological replicates is indicated in parentheses. The median values of Zn and Mn abundance from these experiments are shown in Fig 5.

isolated from *S. pneumoniae* strain D39 and derived Δ*ccnABCDE* strain grown to exponential phase ($OD_{620}$ of ~0.2) at $37^{\circ}$C under an atmosphere of 5% $CO_2$ in BHI broth supplemented with 0.2 mM Zn. Similar to our RNA-seq experiments performed with these strains in the absence of Zn supplementation, we observed a 2.3-fold increase in expression of the CzcD Zn exporter specifying mRNA and a 9.5-fold increase in the Spd_1267 Zn-responsive ECF-type transporter producing mRNA in the Δ*ccnABCDE* mutant compared to its *ccn*+ parent strain (Tables 3 and S5). Additionally, we observed a significant increase in expression of *nrdD* (2.8-fold) and *nrdG* (2.7-fold) encoding the components of the Mn-independent, anaerobic form of ribonucleotide reductase (S5 Table). We validated these results by RT-ddPCR and detected a 2.5-fold and 10.6-fold up-regulation of *czcD* and *spd_1267*, respectively, in the Δ*ccnABCDE* mutant relative to the *ccn*+ D39 strain (Fig 2B and 2C). Interestingly, we also saw a 1.9-fold decrease ($P_{adj}$ = $1.81 \times 10^{-43}$) in expression of the *sodA* mRNA, encoding superoxide dismutase A, in the Δ*ccnABCDE* mutant strain, which was just below our arbitrary two-fold cutoff (S5 Table). We subsequently measured the relative abundance of the *sodA* transcript by northern blot analysis (Fig 6) and consistently observed down-regulation of the *sodA* mRNA when the *ccn* genes were deleted from *S. pneumoniae* during growth in BHI broth supplemented with Zn. This result was intriguing to us since a prior study found that Mn starvation of *S. pneumoniae* cells due to exposure to high concentrations of Zn relative to Mn led to a reduction in transcription of *sodA* and a reduction in superoxide dismutase activity [22]. Furthermore, Eijkelkamp *et* al discovered that deletion of *sodA* had no significant impact on *S. pneumoniae* growth under Mn replete conditions, but was vital for growth in media containing a high Zn-to-Mn ratio [22].

To initially examine whether the growth deficiency of the Δ*ccnABCDE* mutant relative to the *ccn*+ D39 strain was due in part to oxidative stress, we evaluated the impact of addition of Oxyrase, an enzyme mixture that removes molecular oxygen by reducing it to water, on the growth of these strains in BHI broth alone or supplemented with 0.2 mM or 0.4 mM Zn (Fig 7A, 7B, and 7C) under an atmosphere of 5% $CO_2$. Once again, we observed that deletion of *ccnA*, *ccnB*, *ccnC*, *ccnD*, and *ccnE* from *S. pneumoniae* strain D39 had no significant impact on growth in BHI alone. However, under these growth conditions, the addition of Oxyrase reduced the growth rate of both the *ccn*+ and *ccn*- strains to a similar extent (Fig 7A). As we anticipated, addition of Oxyrase to BHI supplemented with Zn (0.2 mM) improved the growth rate of the Δ*ccnABCDE* strain to that observed for the *ccn*+ D39 parent strain (Fig 7B). Interestingly, addition of Oxyrase improved the growth rate of both strains in BHI with 0.4 mM Zn and eliminated any growth differences between them (Fig 7C). Finally, we examined the contribution of *sodA* to the growth of *S. pneumoniae* D39 and derived Δ*ccnABCDE* mutant strain. In BHI broth alone or supplemented with 0.2 mM Zn, deletion of *sodA* reduced the growth rate of the *ccn*+ strain, but did not result in a significant reduction in growth rate of the Δ*ccnABCDE* mutant strain (Fig 8). Based on these results, we concluded that the amount of functional SodA was negligible in the *S. pneumoniae* strain lacking the Ccn sRNAs and thus, deleting *sodA* did not significantly impact its growth, whereas this deletion did impair growth of the isogenic *ccn*+ strain.

## Discussion

High density Tn-seq experiments performed more than a decade ago revealed that sRNAs play a crucial role in regulating *S. pneumoniae* virulence including its ability to colonize the blood, nasopharynx, and lungs of its host [12]. While this discovery in itself may not be surprising, it is astonishing that very little progress has been made towards understanding the functions of these sRNAs given their importance in governing *S. pneumoniae* pathogenesis. Here, we

**Table 3. Genes significantly, differentially expressed between a *S. pneumoniae* D39 and derived Δ*ccnABCDE* strain in both BHI alone or supplemented with Zn[a].**

| D39 locus tag | Gene | Known or predicted function | Fold change (BHI) | Fold change (BHI+Zn) |
|---|---|---|---|---|
| SPD_0025 | | tRNA-specific adenosine-34 deaminase | 84.3 | 49.0 |
| SPD_0027 | *dut* | deoxyuridine 5'-triphosphate nucleotidohydrolase | 3.52 | 3.76 |
| SPD_0028 | | hypothetical protein | 3.80 | 3.02 |
| SPD_0029 | *radA* | DNA repair protein | 3.55 | 2.95 |
| SPD_0080 | *pavB* | cell wall surface anchor family protein | 6.69 | 6.32 |
| SPD_0163 | | DNA binding protein | 2.00 | 2.07 |
| SPD_0222 | *gpmB1* | phosphoglycerate mutase family protein | 25.0 | 21.7 |
| SPD_0243 | *uppS* | undecaprenyl diphosphate synthase | 5.97 | 7.61 |
| SPD_0244 | *cdsA* | phosphatidate cytidylyltransferase | 5.50 | 7.77 |
| SPD_0245 | *eep* | intramembrane protease | 5.57 | 8.59 |
| SPD_0246 | *proS* | prolyl-tRNA synthetase | 5.91 | 9.65 |
| SPD_0247 | *bglA* | 6-phospho-β-glucosidase | 3.57 | 4.73 |
| SPD_0277 | *bglA-1* | 6-phospho-β-glucosidase | 3.53 | 2.73 |
| SPD_0279 | *celB* | cellobiose PTS transporter subunit | 5.05 | 2.97 |
| SPD_0308 | *clpL* | ATP-dependent protease subunit | 13.7 | 9.55 |
| SPD_0350 | *vraT* | cell wall-active antibiotic response protein | 2.19 | 2.62 |
| SPD_0351 | *vraS* | two-component system histidine kinase | 2.29 | 2.74 |
| SPD_0352 | *vraR* | two-component system response regulator | 2.31 | 2.70 |
| SPD_0353 | *alkD* | degenerate DNA alkylation repair enzyme | 2.11 | 2.67 |
| SPD_0354 | *alkD* | degenerate DNA alkylation repair enzyme | 2.37 | 2.69 |
| SPD_0458 | *hrcA* | heat inducible transcription repressor | 3.62 | 3.69 |
| SPD_0459 | *grpE* | heat shock protein | 3.77 | 3.87 |
| SPD_0460 | *dnaK* | protein chaperone | 3.97 | 3.82 |
| SPD_0461 | *dnaJ* | protein chaperone | 3.50 | 3.52 |
| SPD_0474 | *blpZ* | immunity protein | 2.40 | 2.05 |
| SPD_0501 | *licT* | β-glucoside operon antiterminator | 2.91 | 5.26 |
| SPD_0502 | *bglF* | β-glucoside PTS transporter subunit | 3.07 | 4.65 |
| SPD_0503 | *bglA-2* | 6-phospho-β-glucosidase | 2.57 | 3.75 |
| SPD_0537 | | putative Zn-dependent protease | 2.07 | 2.21 |
| SPD_0615 | *glnH3* | degenerate glutamine ABC transporter subunit | 11.6 | 18.0 |
| SPD_0616 | *glnQ3* | glutamine ABC transporter subunit | 8.90 | 16.8 |
| SPD_0617 | *glnP3b* | glutamine ABC transporter subunit | 11.1 | 15.8 |
| SPD_0618 | *glnP3a* | glutamine ABC transporter subunit | 11.8 | 15.1 |
| SPD_0681 | | hypothetical protein | 2.82 | 5.45 |
| SPD_0701 | *ciaR* | two-component response regulator | 2.72 | 2.56 |
| SPD_0702 | *ciaH* | two-component histidine kinase | 2.80 | 3.01 |
| SPD_0775 | | acetyltransferase | 3.29 | 3.61 |
| SPD_0803 | | putative phage shock protein C | | |
| SPD_0804 | | ABC transporter ATP-binding protein | 2.28 | 3.01 |
| SPD_0805 | | ABC transporter permease protein | 2.43 | 3.15 |
| SPD_0913 | | extracellular protein | 3.39 | 3.31 |
| SPD_0938 | | degenerate TN5252 relaxase | 9.35 | 5.06 |
| SPD_0940 | *rrfD* | UDP-N-acetyl-D-mannosaminouronic acid dehydrogenase. | 3.95 | 5.31 |
| SPD_0942 | | hypothetical protein | 2.25 | 2.41 |
| SPD_0943 | | hypothetical proein | 2.41 | 2.43 |
| SPD_0944 | | nodulation protein L | 2.24 | 2.38 |
| SPD_0946 | | hypothetical protein | 2.16 | 3.27 |

*(Continued)*

**Table 3.** (Continued)

| D39 locus tag | Gene | Known or predicted function | Fold change (BHI) | Fold change (BHI+Zn) |
|---|---|---|---|---|
| SPD_0947 | | hypothetical protein | 2.69 | 3.97 |
| SPD_0948 | *nikS* | nikkomycin biosynthesis protein | 3.73 | 4.29 |
| SPD_0949 | | N-acetylneuraminate synthase | 2.38 | 4.85 |
| SPD_0950 | *mefE* | macrolide ABCE transporter subunit | 2.44 | 3.99 |
| SPD_1045 | | degenerate DUF3884 domain protein | 4.73 | 6.81 |
| SPD_1046 | *lacG-2* | 6-phospho-b-galactosidase | 3.56 | 7.28 |
| SPD_1047 | *lacE-2* | lactose PTS transporter subunit | 4.21 | 6.33 |
| SPD_1049 | *lacT* | β-glucoside *bgl* operon antiterminator | 3.23 | 3.48 |
| SPD_1114 | | hypothetical protein | 13.5 | 5.37 |
| SPD_1267 | | ECF transporter subunit | 11.1 | 9.53 |
| SPD_1297 | *pdxS* | pyridoxal 5'-phosphate synthase | 2.02 | 2.04 |
| SPD_1506 | *axe1* | acetyl xylan esterase 1 | 3.62 | 2.68 |
| SPD_1615 | | degenerate hypothetical protein | 4.02 | 2.09 |
| SPD_1638 | *czcD* | Cd/Zn exporter | 2.69 | 2.33 |
| SPD_1709 | *groL* | HSP60 family chaperone | 2.58 | 2.38 |
| SPD_1710 | *groES* | HSP60 family chaperone | 2.25 | 2.31 |
| SPD_1716 | | hypothetical protein | 2.56 | 5.62 |
| SPD_1717 | | membrane protein | 2.40 | 5.22 |
| SPD_1718 | | LytR/AlgR family response regulator | 2.44 | 4.58 |
| SPD_1746 | | hypothetical protein | 2.96 | 4.25 |
| SPD_1747 | *pneA1* | lantibiotic peptide | 2.02 | 4.29 |
| SPD_1748 | *pneA2* | lantibiotic peptide | 2.19 | 4.60 |
| SPD_1749 | *lanM* | lanthionine biosynthesis protein | 2.49 | 2.37 |
| SPD_1750 | *wrbA* | FAD-dependent flavoprotein | 3.00 | 2.85 |
| SPD_1751 | | hypothetical protein | 2.56 | 4.07 |
| SPD_1752 | *clyB* | toxin secretion ABC transporter | 3.58 | 4.02 |
| SPD_1753 | | epidermin leader peptide processing serine protease | 2.44 | 3.00 |
| SPD_1769 | | membrane protein | 2.29 | 3.42 |
| SPD_1932 | *malP* | malodextrin phosphorylase | 2.58 | 2.95 |
| SPD_1933 | *malQ* | 4-α-glucanotransferase | 2.76 | 2.77 |
| SPD_1990 | | mannose PTS transporter subunit | 2.01 | 13.8 |
| SPD_1994 | *fucA* | L-fuculose phosphate aldolase | 2.35 | 8.69 |
| SPD_2034 | *comFC* | phosphororibosyltransferase domain protein | 32.7 | 14.1 |
| SPD_2035 | *comFA* | DNA transporter ATPase | 8.88 | 10.2 |
| SPD_2068 | *htrA* | serine protease | 2.79 | 2.13 |
| SPD_2069 | *parB* | chromosome partitioning protein | 3.04 | 2.88 |

[a]RNA extraction and mRNA-seq analyses were performed as described in *Materials and Methods*. RNA was prepared from cultures of strains IU1781 (D39 *rpsL1*) and NRD10176 (D39 *rpsL1 ΔccnABCDE*) grown to exponential phase in BHI alone or supplemented with 0.2 mM ZnSO$_4$ (S1 and S2 Tables). Fold changes (2.0-fold cut-off) and adjusted P-values (Pval <0.05) are based on three independent biological replicates.

investigated the contribution of the five homologous Ccn sRNAs to *S. pneumoniae* pathogenesis and gene regulation. In addition to confirming their crucial role in pneumococcal disease progression (Fig 1), we have discovered their extensive functions in regulating gene expression and Zn resistance. The Zn sensitivity of *S. pneumoniae* strains lacking the Ccn sRNA genes likely contributes to their reduced virulence.

Zn is an important transition metal for *S. pneumoniae* during host infection. Prior work has shown that *S. pneumoniae* requires Zn for pathogenesis as deletion of genes for the Zn

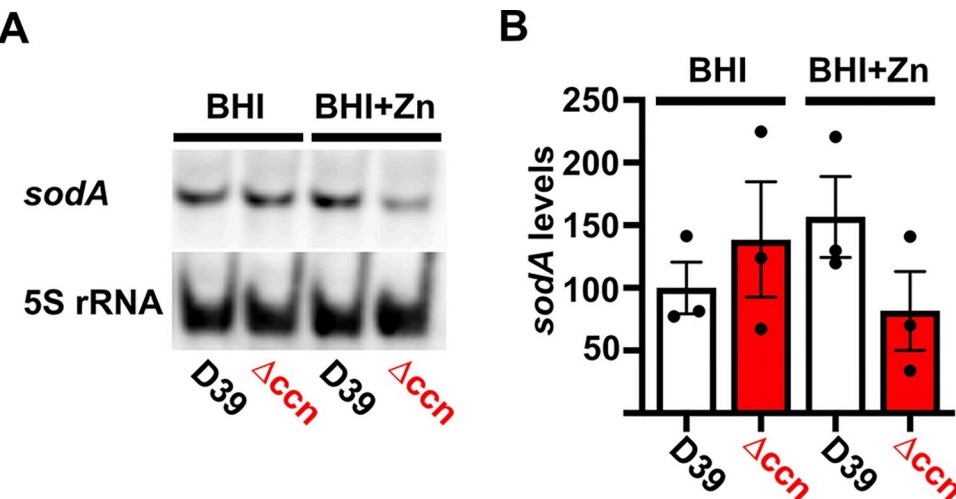

**Fig 6. Effects of the *ccn* genes on expression of *sodA* mRNA.** Levels of the *sodA* mRNA were determined by northern blot analyses as described in Materials and Methods for strain IU1781 (D39) and derived Δ*ccnABCDE* mutant strain (NRD10176; Δ*ccn*) grown to exponential phase ($OD_{620}$ of ~0.2) in BHI broth alone (BHI) or supplemented with 0.2 mM $ZnSO_4$ (BHI+Zn) at 37°C under an atmosphere of 5% $CO_2$. Representative blots are shown in (A). Levels of *sodA* mRNA normalized to 5S rRNA abundance are presented in (B). Values represent the mean of three independent cultures and error bars indicate SEM.

binding components of its only known Zn acquisition system (*adcA* and *adcAII*) abolished pneumococcal virulence in murine nasopharyngeal colonization, septicemia, and pneumonia models [30] and reduced pneumococcal burden in lungs, pleural cavity, and blood of mice fed a Zn depleted or replete diet [31]. However, previous studies have also established that *S. pneumoniae* must combat high Zn levels during host infection as deletion of the gene for its only known Zn exporter (*czcD*) also significantly reduced pneumococcal burden in the lungs and blood of mice fed a Zn replete diet following intranasal infection [31]. In that study, Zn levels were shown to increase in the blood, lungs, and nasopharynx of mice following infection with the pneumococcus, and the areas in which Zn were most abundant were regions containing pneumococcal cells [31]. Finally, experiments showing that deletion of *czcD* from *S. pneumoniae* renders it susceptible to killing by macrophage-like cells derived from human Thp-1 cells [31] indicate that Zn is used by phagocytic cells to poison *S. pneumoniae*.

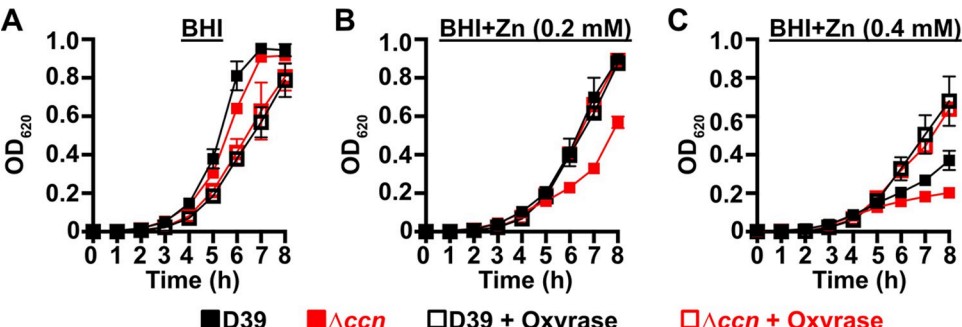

**Fig 7. Reduction of $O_2$ abolishes the Zn hypersensitivity of the *S. pneumoniae* Δ*ccnABCDE* mutant.** Growth characteristics at 37°C under an atmosphere of 5% $CO_2$ in BHI broth alone (A) or with 0.2 mM (B) or 0.4 mM (C) $ZnSO_4$ of IU781 (D39) and NRD10176 (Δ*ccn*) in the absence or presence of 10% (volume/volume) Oxyrase. Each point on the graph represents the mean $OD_{620}$ value from three independent cultures. Error bars, which in some cases are too small to observe in the graph, represent the standard deviation (SD).

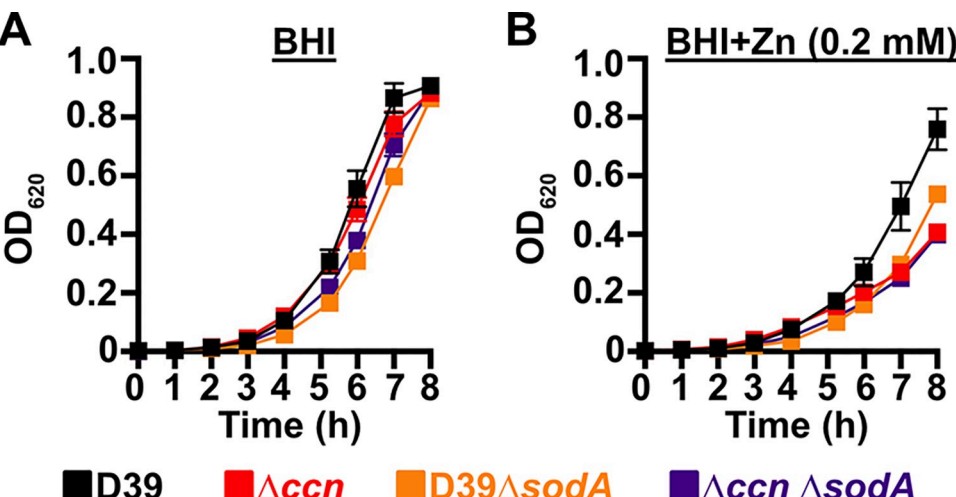

**Fig 8. A *S. pneumoniae* Δ*sodA* mutant phenocopies the Zn hypersensitivity of a Δ*ccnABCDE* mutant strain.**
Growth characteristics at 37°C under an atmosphere of 5% $CO_2$ in BHI broth alone (A) or with 0.2 mM $ZnSO_4$ (B) of IU781 (D39), NRD10176 (Δ*ccn*), NRD10533 (D39 Δ*sodA*), and NRD10534 (Δ*ccn* Δ*sodA*). Each point on the graph represents the mean $OD_{620}$ value from three independent cultures. Error bars, which in some cases are too small to observe in the graph, represent the standard deviation (SD).

In our work presented here, we found that exposure to relatively high, yet host-relevant, Zn concentrations (0.2 mM) disproportionally inhibited *S. pneumoniae* Δ*ccnABCDE* mutant strain growth in BHI broth (Fig 3, 4, 7, and 8) and increased total cell-associated Zn levels (Fig 5 and Table 2) of *S. pneumoniae* strains lacking genes for the five Ccn sRNAs grown in C-medium alone or supplemented with 0.2 mM $ZnSO_4$. 0.2 mM is a Zn concentration comparable to what was found in the nasopharynx of mice infected with the pneumococcus, but far lower than its abundance in blood (~0.6 mM). These results suggest that the Zn sensitivity caused by loss of the *ccn* genes likely contributes to the reduced virulence of *S. pneumoniae* Δ*ccnABCDE* mutant strains. While we were unable to detect by ICP-MS a statistically significant difference in the Zn content between *S. pneumoniae* and derived Δ*ccnABCDE* mutant strain grown in BHI broth alone or supplemented with 0.2 mM Zn, we suspect that loss of the Ccn sRNA genes does increase the amount of bioavailable Zn in *S. pneumoniae* under these growth conditions since increased *czcD* expression is a sensitive indicator of high levels of bioavailable Zn inside of pneumococcal cells [24].

Prior work established that Zn inhibits Mn uptake by *S. pneumoniae* [26] and compromises the ability of *S. pneumoniae* to defend itself from oxidative stress due to inhibition of superoxide dismutase A (SodA) activity, when Zn is far more abundant in its environment than Mn. We tested whether the Zn sensitivity caused by deletion of the *ccn* genes from *S. pneumoniae* was due to a defect in Mn homeostasis and its oxidative stress response. In short, we found that the Zn-dependent growth inhibition caused the *ccnABCDE* deletion was completely alleviated by addition of Mn (Fig 4) or Oxyrase (Fig 7), which removes molecular oxygen by reducing it to water. Furthermore, deletion of *sodA*, encoding the Mn-dependent superoxide dismutase A, from *S. pneumoniae* resulted in a Zn-dependent growth inhibition; however, the same deletion had no impact on the growth of the Δ*ccnABCDE* strain (Fig 8). Altogether, these results indicate that the Ccn sRNAs promote Zn homeostasis resulting in an increased abundance of active SodA, which improves the growth of *S. pneumoniae* in a Zn-rich environment due to greater protection from damaging reactive oxygen species.

How do the Ccn sRNAs prevent *S. pneumoniae* from accumulating bioavailable $Zn^{2+}$ cations relative to $Mn^{2+}$? The Ccn sRNAs could preclude a Zn buildup by (1) promoting

expression of a Zn exporter, (2) negatively regulating expression of a Zn importer, or (3) increasing production of an intracellular protein or other factor that effectively chelates Zn. As mentioned already, CzcD is the main Zn exporter in *S. pneumoniae* and is essential for Zn resistance [32]. In contrast, the Adc system is important for Zn uptake, but supplementation with Zn is able to bypass the requirement for this transporter indicating that at least one unidentified low-affinity Zn importer exists in *S. pneumoniae* [30,33,34]. Our global analysis of gene expression in *S. pneumoniae* D39 or TIGR4 and derived Δ*ccnABCDE* mutant strains revealed that *czcD* expression increased when the *ccn* genes were removed, whereas no significant difference in expression of any of the *adc* genes was observed. Thus, while we are not able to rule out the possibility that the Ccn sRNAs regulate expression of an uncharacterized transporter capable of translocating Zn, our results indicate that the Zn sensitivity of the Δ*ccnABCDE* mutant strain is not due to reduced expression of *czcD* or up-regulation of the Adc system. While it also remains possible that the Ccn sRNAs regulate production of an unknown factor that chelates or chaperones intracellular Zn, we did not observe a reduction in the expression of any *known* Zn-binding proteins in the *ccn* mutant relative to its parental *ccn*+ *S. pneumoniae* D39 or TIGR4 strain via RNA-seq (Tables 1, 3, S3, S4, and S5).

Even though, we did not observe a statistically significant decrease in total cell-associated Mn in the Δ*ccnABCDE* mutant relative to its parental *S. pneumoniae* D39 strain, we still wondered whether or not loss of the Ccn sRNAs caused Zn sensitivity due to reduced uptake or increased export of Mn, since the ICP-MS based approach that we utilized measures total, not bioavailable metal abundance. The main Mn exporter of *S. pneumoniae* is MntE, as deletion of the encoding gene leads to accumulation of total cell associated Mn [25,35]. MgtA, designated as a Ca efflux protein, appears to also export Mn, but has a very limited role in this process [36]. Neither MntE or MgtA were up-regulated in either *S. pneumoniae* strain D39 or TIGR4 when the *ccn* genes were deleted (S3, S4, and S5 Tables) making it unlikely that the Ccn sRNAs increase intracellular Mn levels by down-regulating expression of these Mn exporter genes. Additionally, we were unable to identify strong Ccn sRNA binding sites in the translation initiation region of *mntE* or *mgtA*, which suggests that these sRNAs do not directly regulate translation of these transcripts. Finally, if the Ccn sRNAs increase total cell-associated Mn levels by down-regulating MntE expression, then we would expect that deletion of *mntE* would suppress the Zn hypersensitivity of the *S. pneumoniae* Δ*ccnABCDE* mutant; however, this did not occur (S6A and S6C Fig).

An alternative possibility is that the Ccn sRNAs promote Mn uptake by positively regulating expression of the *psaBCA* operon encoding the only known Mn importer in *S. pneumoniae* [33, 37]. Localized to the inner membrane, PsaB is the ATP binding component whereas PsaC is the permease of this ABC-type transporter. PsaA, the substrate binding component, is located in the periplasm, where it binds Mn. Once again, in our RNA-seq experiments, we did not observe a decrease in expression of the *psaBCA* operon when the *ccn* genes were deleted from *S. pneumoniae* strain D39 or TIGR4 (S3, S4, and S5 Tables) indicating that the Ccn sRNAs do not positively regulate expression of this Mn importer. Furthermore, if this was the case, then we would expect that deletion of *psaR* encoding the repressor of the *psaBCA* operon [38, 39] might suppress the Zn-dependent growth inhibition of the *S. pneumoniae* Δ*ccnABCDE* mutant; however, we did not observe this (S6B and S6D Fig).

In summary, we show that Ccn sRNAs play a key role in controlling the ability of *S. pneumoniae* to cause invasive pneumonia (Fig 1) and resist Zn intoxication (Fig 3). Our results indicate that the reduced growth of *S. pneumoniae* in the presence of excess, but physiologically relevant Zn concentrations caused by loss of the Ccn sRNA is due to an increase in oxidative stress (Figs 7 and 8). Our work suggests that there are likely additional, uncharacterized factors that modulate bioavailable Zn abundance in pneumococcus.

## Materials and Methods

### Ethics statement

All animal procedures were performed at the University of Texas Health Science Center at Houston with prior approval by University of Texas Health Science Center Animal Welfare Committee. The health and well-being of all laboratory animals were overseen by the Center for Laboratory Animal Medicine and Care (CLAMC). The University of Texas Health Science Center Animal Care and Use Program is fully accredited by the Association for Assessment and Accreditation of Laboratory Animal Care (AAALAC).

### Bacterial strains and growth conditions

Bacterial strains used in this study were derived from encapsulated *S. pneumoniae* serotype 2 strain D39W [14] and TIGR4 and are listed in S1 Table. Strains were grown on plates containing trypticase soy agar II (modified; Becton-Dickinson [BD]) and 5% (vol/vol) defibrinated sheep blood (TSAII BA) at 37°C in an atmosphere of 5% $CO_2$, and liquid cultures were statically grown in BD brain heart infusion (BHI) broth or C-medium [40] at 37°C in an atmosphere of 5% $CO_2$. C-medium was prepared as described by Lacks and Hotchkiss, but water was added in place of yeast extract. Bacteria were inoculated into BHI broth from frozen cultures or single, isolated colonies. For overnight cultures, strains were first inoculated into a 17-mm-diameter polystyrene plastic tube containing 5 mL of BHI broth and then serially diluted by 100-fold into four tubes; these cultures were then grown for 10 to 16 h. Cultures with an optical density at 620 nm ($OD_{620}$) of 0.1 to 0.4 were diluted to a starting $OD_{620}$ between 0.002 and 0.005 in 5 mL of BHI broth in 16-mm glass tubes. For growth in C-medium, 2 mL of overnight cultures grown in BHI with an $OD_{620}$ of 0.1 to 0.4 were spun down at 21,000 x $g$ for 2.5 min at room temperature. The supernatant was removed, and the pellet was washed with 1.0 mL of C-medium. The solution was vortexed to resuspend the pellet and spun again at 21,000 x $g$ for 2.5 min at room temperature. The supernatant was removed and the pellet was resuspended in 4.0 mL of C-medium. $OD_{620}$ was used to determine how much culture to add to 5.0 mL of C-medium in 16 mm glass tubes to begin growth at $OD_{620}$ = 0.002. Growth was monitored by measuring $OD_{620}$ using a Genesys 30 visible spectrophotometer (ThermoFisher Scientific). For antibiotic selections, TSAII BA plates and BHI cultures were supplemented with 250 μg kanamycin per mL, 150 μg streptomycin per mL, or 0.3 μg erythromycin per mL.

### Construction and confirmation of mutants

Mutant strains were constructed by transformation of competent *S. pneumoniae* D39 and TIGR4 derived strains with linear PCR amplicons as described previously [41,42]. DNA amplicons containing antibiotic resistance markers were synthesized by overlapping fusion PCR using the primers listed in S2 Table. Competence was induced in *S. pneumoniae* D39 or TIGR4 derived cells with CSP-1 or CSP-2, respectively, synthetic competence stimulatory peptide. Unmarked deletions of the target genes were constructed using the *kan*$^R$-*rpsL*$^+$ (Janus cassette) allele replacement method as described previously [43]. In the first step, the Janus cassette containing *rpsL*$^+$ allele and a kanamycin resistance gene was used to disrupt target genes in an *rpsL1* or *rpsLK56T* (Str$^R$) strain background, and transformants were selected for kanamycin resistance and screened for streptomycin sensitivity. In the second step, the Janus cassette was eliminated by replacement with a PCR amplicon lacking antibiotic markers and the resulting transformants were selected for streptomycin resistance and screened for kanamycin sensitivity. Freezer stocks were made of each strain from single colonies isolated twice

on TSAII BA plates containing antibiotics listed in S1 Table. All strains were validated by PCR amplification and sequencing.

## RNA extraction

To isolate RNA, strains were grown in 30 mL of BHI starting at an $OD_{620}$ = 0.002 in 50 mL conical tubes. RNA was extracted from exponentially growing cultures of IU1781 (D39), NRD10220 (TIGR4), and their derived isogenic mutants lacking all five *ccn* genes, NRD10176 (D39 Δ*ccn*) and NRD10266 (TIGR4 Δ*ccn*), at $OD_{620}$ ≈ 0.2 using the FastRNA Pro Blue Kit (MP Bio) according to the manufacturer's guidelines. Briefly, cells were collected by centrifugation at 16,000 x g for 8 min at 4°C. Cell pellets were resuspended in 1 mL of RNApro solution (MP Bio) and processed five-times for 40 sec at 400 rpm in a BeadBug homogenizer (Benchmark Scientific). Cell debris was removed by centrifugation at 16,000 x g for 5 min at 4°C. After mixing 300 μL of chloroform with the supernatant, the aqueous and organic layers were separated by centrifugation at 16,000 x g for 5 min at 4°C. RNA was precipitated with 500 μL of ethanol at -80°C overnight. After collecting the precipitated RNA by centrifugation at 16,000 x g for 15 min at 4°C, the pellet was washed once with 75% ethanol and suspend in DEPC-treated water. The amount and purity of all RNA samples isolated were assessed by NanoDrop spectroscopy (Thermo Fisher).

## Library preparation and mRNA-seq

cDNA libraries were prepared from total RNA Azenta Life Sciences. Briefly, total RNA was subjected to rRNA-depletion using the FastSelect 5S/16S/23S rRNA depletion kit for bacteria. Libraries were the generated with NEBNext Ultra II Directional RNA Library Prep Kit. 150 bp paired-end read sequencing was performed using an Illumina HiSeq4000 sequencer.

## RNA-seq analysis

The raw sequencing reads were quality and adapter trimmed using Cutadapt version 4.1 with a minimum length of 18 nucleotides. The trimmed reads were then mapped on the *Streptococcus pneumoniae* D39 (Genbank CP000410) genome using Bowtie2 [44]. HTseq version 2.0.2 was used to generate read counts for the genes [45]. Differential gene expression was identified using the program DESeq2 with default parameters [46]. Primary data from the mRNA-seq analyses were submitted to the NCBI Gene Expression Omnibus (GEO) and have the accession number GSE246655.

## Reverse transcriptase-droplet digital PCR (RT-ddPCR) analysis

RT-ddPCR was performed as described previously [47]. Isolated RNA was treated with DNase (TurboDNase, Ambion) as per manufacturer's instructions. Next, RNA (1 μg) was reverse transcribed using Superscript III reverse transcriptase (Invitrogen) with random hexamers. RT and No RT control (NRT) sample were utilized. These samples were diluted $1:10^1$, $1:10^2$, $1:10^3$, $1:10^4$ or $1:10^6$. Then, 2 μL of each diluted RT and NRT sample was added to a 22 μL reaction mixture containing 11 μL of QX200 ddPCR Evagreen Supermix (Bio-Rad) and 1.1 μL of each 2 μM ddPCR primers (S6 Table). A single no template control (NTC) was included for each ddPCR primer pair used. Reactions were performed using at least three independent biological replicates. Droplets were generated using the QX200 Automated Droplet Generator (Bio-Rad), and end-point PCR was carried out using a C1000 Touch thermal cycler (Bio-Rad) following the manufacturer's instructions. Quantification of PCR-positive and PCR-negative droplets in each reaction, which provides absolute quantification of the target transcript, was

performed using the QX200 Droplet Reader (Bio-Rad). This data was analyzed with Quanta-Soft software (Bio-Rad) to determine the concentration of each target expressed as copies per μL. Transcript copies were normalized to *tuf* mRNA (internal control) and fold changes of transcripts corresponding to target genes in different mutants relative to the WT parent were calculated. Statistical analysis was performed using Student's t-test with GraphPad Prism version 10.0.0.

## Northern blot analysis

Northern blotting was conducted as previously described [13]. Briefly, 3 μg of isolated RNA was fractionated on 10% polyacrylamide gels containing 7% urea by electrophoresis at 55 V and subsequently, transferred to a Zeta-probe membrane (Bio-Rad) using a Trans-Blot SD semidry transfer apparatus (Bio-rad) at 4 mA per $cm^2$ with a maximum of 400 mA for 50 min. RNA was then UV-crosslinked to the membrane with a Spectroline UV crosslinker with the "optimal crosslink" setting. 5'-Biotinylated probes were hybridized to the membrane overnight at 42°C in ULTRAhyb (Ambion) hybridization buffer. Blots were developed according to the BrightStar BioDetect kit protocol (Ambion), imaged with the ChemiDoc MP imager (Bio-Rad), and individual band intensities were quantified using Image Lab software version 5.2.1 (Bio-Rad). Signal intensities for each transcript were normalized to that of 5S rRNA, which served as a loading control. Graphs of normalized abundance of each transcript for three biological replicates were produced using GraphPad Prism version 10.0.0.

## Inductively coupled plasma-mass spectrometry (ICP-MS) analysis

ICP-MS sample preparation was based on a previous publication [48], with some modifications. Metal-free microfuge tubes were used throughout, and pipette tips were rinsed prior to use. Bacteria were grown in BHI broth or C medium at 37°C with 5% $CO_2$ to $OD_{620}$ = 0.2. Five mL of culture was centrifuged for 10 min in pre-chilled tubes at 12,400 x $g$ at 4°C, and cell pellets were resuspended in 1.0 mL of chilled BHI supplemented with 1 mM nitrilotriacetic acid (Sigma-Aldrich) (pH 7.2). Samples were centrifuged for 7 min at 16,100 x $g$ at 4°C, and supernatants were removed. Pellets were centrifuged for an additional 3 min in the same way, and residual supernatant was removed. Cell pellets were washed twice with centrifugation in the same way with 1.0 mL of chilled PBS lacking $K^+$ (130 mM NaCl, 8.8mM $Na_2HPO_4$, 1.2mM $NaH_2PO_4$, pH 7.0) that had been treated with chelator. Chelated PBS was prepared by mixing with 1% (wt/vol) Chelex-100 (BioRad), which was rotated overnight at 4°C and passed through a 0.22 μm Steriflip (MilliporeSigma) filter. Before the last centrifugation in PBS, samples were split into two 0.475 mL aliquots for ICP-MS analysis and protein quantification. After removal of supernatants, pellets for ICP-MS were dried for 15 h at low heat in an evaporative centrifuge and stored at -80°C until being processed for ICP-MS analysis. Pellets for protein determination were suspended in in 100 μL of lysis buffer (1% (wt/vol) SDS [Sigma], 0.1% w/v Triton X-100 [Mallinckrodt]) and stored at -80°C. Protein amount was determined by using the DC protein assay (BioRad). For ICP-MS analysis, dried samples were resuspended in 400 μL of 30% trace metal grade $HNO_3$ (Sigma). Samples and a 30% $HNO_3$ blank were heated at 95°C for 10 min with shaking at 500 rpm. Samples were then diluted 100-fold to a final volume of 3.0 mL with 2.5% $HNO_3$ containing the Pure Plus Internal Standard Mix (100 ppb, PerkinElmer). Samples were analyzed using an Agilent 8800 QQQ ICP-MS operating with hydrogen ($^{55}$Mn detection) or helium ($^{66}$Zn detection) as collision gases to remove possible interferences. $^{45}$Sc or $^{72}$Ge were used as internal references. $Zn^{2+}$ and $Mn^{2+}$ amounts were calculated from standard curves made with Pure Plus Multi-Element Calibration Standard 3 (0.5-100ppb, PerkinElmer). Metals amounts detected in the 30% $HNO_3$ blank were subtracted

from all samples. Metal amounts in samples were normalized relative to total protein amounts in the matched samples.

## Mouse models of infection

All procedures were approved in advance by University of Texas Health Science Center Animal Welfare Committee and carried out as previously described [47]. Male ICR mice (21–24 g; Envigo) were anaesthetized by inhaling 4 to 5% isoflurane. A total of 8 mice were intranasally inoculated with $10^7$ CFU of a specific *S. pneumoniae* strain suspended in 50 μL of 1 X PBS prepared from cultures grown in BHI broth at 37$^°$C in an atmosphere of 5% $CO_2$ to $OD_{620} \approx 0.1$. Mice were monitored visually at 4 to 8 h intervals, and isoflurane-anesthetized moribund mice were euthanized by cardiac puncture-induced exsanguination followed by cervical dislocation. Kaplan-Meir survival curves and log-rank tests were generated using GraphPad Prism 10.0.0 software.

## Supporting information

**S1 Fig. Virulence phenotypes of *S. pneumoniae* strains harboring deletion of individual *ccn* genes.** Survival curve of ICR outbred mice after infection with ~$10^7$ CFU in a 50 μL inoculum of the following *S. pneumoniae* strains: (A) IU781 (D39), NRD10073 (Δ*ccnA*), and NRD10077 (Δ*ccnE*); (B) IU781 (D39), NRD10074 (Δ*ccnB*), NRD10075 (Δ*ccnC*), and NRD10076 (Δ*ccnD*). Eight mice were infected per strain. Disease progression of animals was monitored, the time at which animals reached a moribund state was recorded, and these mice were subsequently euthanized as described in Materials and Methods. A survival curve was generated from this data and analyzed by Kaplan-Meier statistics and log rank test to determine P-values. (TIF)

**S2 Fig. Growth phenotypes of *S. pneumoniae* D39 derived strains harboring deletion of specific *ccn* genes.** Growth characteristics at 37$^°$C under an atmosphere of 5% $CO_2$ in BHI broth alone (A,C, E, G, I, K, M, O) or with 0.2 mM $ZnSO_4$ (B, D, F, H, J, L, N, P) of the following strains: (A, B) IU781 (D39), NRD10073 (Δ*ccnA*), and NRD10074 (Δ*ccnB*); (C, D) IU781 (D39), NRD10075 (Δ*ccnC*), NRD10076 (Δ*ccnD*), and NRD10077 (Δ*ccnE*); (E, F) IU781 (D39), NRD10165 (Δ*ccnACE*), and NRD10166 (Δ*ccnADE*); (G, H) IU781 (D39), NRD10376 (Δ*ccnBCE*), NRD10379 (Δ*ccnBDE*), and NRD10380 (Δ*ccnCDE*); (I, J) IU781 (D39), NRD10081 (Δ*ccnBCD*), and NRD10084 (Δ*ccnACD*); (K, L) IU781 (D39), NRD10372 (Δ*ccnABC*), NRD10373 (Δ*ccnABD*), and NRD10374 (Δ*ccnABE*); (M, N) IU781 (D39), NRD10085 (Δ*ccnBCDE*), and NRD10174 (Δ*ccnACDE*); (O,P) IU781 (D39), NRD10172 (Δ*ccnABCE*), NRD10173 (Δ*ccnABDE*), and NRD10175 (Δ*ccnABCD*). Each point on the graph represents the mean $OD_{620}$ value from three independent cultures. Error bars, which in some cases are too small to observe in the graph, represent the standard deviation (SD). (TIF)

**S3 Fig. Growth phenotypes of *S. pneumoniae* D39, Δ*ccnABCDE* mutant, and derived strain complemented with *ccnC* and *ccnD* or *ccnA* and *ccnB*.** Growth characteristics at 37$^°$C under an atmosphere of 5% $CO_2$ in BHI broth alone (A, C) or with 0.2 mM $ZnSO_4$ (B, D) of IU1781 (D39), NRD10176 (Δ*ccn*), and NRD10397 (Δ*ccn*//*ccnC*$^+$*D*$^+$) (A,B) or NRD10393 (Δ*ccn*//*ccnA*$^+$*B*$^+$) (C, D). Each point on the graph represents the mean $OD_{620}$ value from three independent cultures. Error bars, which in some cases are too small to observe in the graph, represent the standard deviation (SD). (TIF)

**S4 Fig. Growth phenotypes of *S. pneumoniae* TIGR4 derived strains harboring deletion of the *ccn* genes.** Growth characteristics at 37$^°$C under an atmosphere of 5% $CO_2$ in BHI broth

alone (A) or with 0.2 mM (B) or 0.4 mM (C) $ZnSO_4$ of NRD10311 (TIGR4; TIGR4 $rpsL^+$-$rpsG^+$-$cat$) and NRD10346 ($\Delta ccn$; TIGR4 $rpsL^+$-$rpsG^+$-$cat$ $\Delta ccnABCDE$). Each point on the graph represents the mean $OD_{620}$ value from three independent cultures. Error bars, which in some cases are too small to observe in the graph, represent the standard deviation (SD). (TIF)

**S5 Fig. Doubling times of *S. pneumoniae* D39, Δ*ccnABCDE* mutant, or Δ*ccnABCDE* mutant strain complemented with *ccnA*, *ccnB*, and *ccnC* in C medium alone or supplemented with Zn.** Shown are the mean doubling times during exponential growth of IU1781 (D39), NRD10176 (Δ*ccn*), and NRD10396 (Δ*ccn*//*ccnA*$^+$*B*$^+$*D*$^+$) grown in C medium alone or supplemented with 0.2 mM ZnS04 as described in *Materials and Methods*. Doubling times for individual replicates are shown with solid lines indicating the mean of six different biological replicates and error bars denoting standard error of the mean (SEM). Statistical significance as determined by a Mann-Whitney test is indicate as * ($P < 0.05$), ** ($P < 0.005$), *** ($P < 0.0005$), or **** ($P < 0.00005$). (TIF)

**S6 Fig. Growth phenotypes of *S. pneumoniae* D39 and derived strains harboring deletion of the *ccn* genes and/or *psaR* or *mntE*.** Growth characteristics at 37$^{\circ}$C under an atmosphere of 5% $CO_2$ in BHI broth alone (A, B) or with 0.2 mM $ZnSO_4$ (C, D) of the following strains: (A, C) IU781 (D39), NRD10176 (Δ*ccnABCDE*), NRD10448 (Δ*mntE*), and NRD10450 (Δ*ccnABCDE* Δ*mntE*); (B, D) IU781 (D39), NRD10176 (Δ*ccnABCDE*), NRD10447 (Δ*psaR*), and NRD10450 (Δ*ccnABCDE* Δ*psaR*). Each point on the graph represents the mean $OD_{620}$ value from three independent cultures. Error bars, which in some cases are too small to observe in the graph, represent the standard deviation (SD). (TIF)

**S1 Table. *S. pneumoniae* strains used in this study.** (DOCX)

**S2 Table. Primers used to construct mutants used in this study.** (DOCX)

**S3 Table. Comparison of gene expression *S. pneumoniae* D39 and derived Δ*ccnABCDE* strains in BHI broth by RNA-seq.** (XLSX)

**S4 Table. Comparison of gene expression *S. pneumoniae* TIGR4 and derived Δ*ccnABCDE* strains in BHI broth by RNA-seq.** (XLSX)

**S5 Table. Comparison of gene expression *S. pneumoniae* D39 and derived Δ*ccnABCDE* strains in BHI broth with 0.2 mM $ZnSO_4$ by RNA-seq.** (XLSX)

**S6 Table. Oligonucleotide primers used for qRT-PCR.** (DOCX)

## Author Contributions

**Conceptualization:** Nicholas R. De Lay, Dhriti Sinha, Abigail Garrett, David P. Giedroc, Malcolm E. Winkler.

**Data curation:** Nicholas R. De Lay.

**Formal analysis:** Nicholas R. De Lay, Nidhi Verma, Dhriti Sinha, Abigail Garrett.

**Funding acquisition:** Nicholas R. De Lay, Maximillian K. Osterberg, David P. Giedroc, Malcolm E. Winkler.

**Investigation:** Nicholas R. De Lay, Nidhi Verma, Dhriti Sinha, Abigail Garrett, Maximillian K. Osterberg, Daisy Porter, Spencer Reiling.

**Methodology:** Nicholas R. De Lay.

**Project administration:** Nicholas R. De Lay.

**Resources:** Nicholas R. De Lay, David P. Giedroc, Malcolm E. Winkler.

**Supervision:** Nicholas R. De Lay, David P. Giedroc, Malcolm E. Winkler.

**Writing – original draft:** Nicholas R. De Lay.

**Writing – review & editing:** Nicholas R. De Lay, David P. Giedroc, Malcolm E. Winkler.

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
