## [Decision Letter · Decision Letter 0]

3 Jun 2024

Dear Dr. De Lay,

Thank you very much for submitting your manuscript "The five homologous CiaR-controlled Ccn sRNAs of Streptococcus pneumoniae modulate Zn-resistance." for consideration at PLOS Pathogens. As with all papers reviewed by the journal, your manuscript was reviewed by members of the editorial board and by several independent reviewers. In light of the reviews (below this email), we would like to invite the resubmission of a significantly-revised version that takes into account the reviewers' comments.

The inclusion of the complete set of strains as requested by 2 reviewers in the experiments shown in Figure 1 is essential along with an extended duration of the growth kinetics. An analysis of czcD mRNA expression with or without high levels of zinc would address the question about zinc dependence of regulation. Finally, the question about analysis of bacterial burdens in the blood and tissue of wild type vs mutant systemic infections could highlight important aspects about the role of these sRNAs during in vivo pathogenesis beyond survival curves.

We cannot make any decision about publication until we have seen the revised manuscript and your response to the reviewers' comments. Your revised manuscript is also likely to be sent to reviewers for further evaluation.

Sincerely,

Helena Ingrid Boshoff

Section Editor

PLOS Pathogens

Helena Boshoff

Section Editor

PLOS Pathogens

Michael Malim

Editor-in-Chief

PLOS Pathogens

orcid.org/0000-0002-7699-2064

The inclusion of the complete set of strains as requested by 2 reviewers in the experiments shown in Figure 1 is essential along with an extended duration of the growth kinetics. An analysis of czcD mRNA expression with or without high levels of zinc would address the question about zinc dependence of regulation. Finally, the question about analysis of bacterial burdens in the blood and tissue of wild type vs mutant systemic infections could highlight important aspects about the role of these sRNAs during in vivo pathogenesis beyond survival curves.

Reviewer's Responses to Questions

**Part I - Summary**

Reviewer #1: This manuscript by De Lay et. al. studies the impact of the five regulatory Ccn RNAs on the sensitivity of S. pneumoniae to the trace metal zinc. To this end, the authors employed growth assays with various mutants and metal conditions, a transcriptomic approach and mammalian infection model to investigate the impact of the Ccn sRNAs. Overall, this is an interesting manuscript, although the exact mechanisms of how the Ccn sRNAs regulate their putative target genes and how they impair zinc toxicity are not completely solved. Major and minor comments are listed below.

Reviewer #2: In this work, De Lay et al., provide evidence that 5 sRNAs from S. pneumoniae contribute to systemic infection and are involved in the regulation of zinc stress resistance. They show that mutation of all 5 sRNAs, but not mutation of individual sRNAS, results in increased sensitivity to zinc toxicity, and that this is likely to due to an imbalance of intracellular Zn:Mn that could impact downstream function of superoxide dismutase and result in a sensitivity to oxidative stress. This work provides exciting new information of potential mechanisms of sRNA regulation and the impact on bacterial homeostasis and virulence. This manuscript would benefit from adding more data obtained from the animal studies about bacterial burden – this was very briefly mentioned in the text in the context of the blood, but it would be nice to see tissue data as well. The data presented are interesting, but I found the formatting for the figures and the results to be a bit chaotic and required scanning back-and-forth from different sections - overall, this made the manuscript confusing and led to the feeling of redundancy as you read through the text.

Reviewer #3: This well-written manuscript by De Lay and colleagues addresses an important gap in the literature: the role of sRNA’s in Spn and Gram-positive bacteria overall. The study focuses on five homologous sRNAs (ccnA-E) and demonstrates their role in Zn resistance. Using a genetic approach, the study shows that these sRNA influence the ratio of Zn to Mn, and that this ratio is more prone to increase to toxic levels when ccns are deleted. In agreement, they show that defects in growth linked the deletion of ccn with extra Zn, are rescued when Mn is added to the growth media. Further, guided by expression data, they demonstrate that these sRNA alleviate oxidative stress and that in wild type cells sodA also participates in this response. Finally, they show that these sRNAs promote virulence in a murine model of pneumococcal invasive disease. Further, the work uses two strains, D39 and TIGR4, for its discovery and validation suggesting the findings are not strain-specific. Overall, the paper is an important contribution to the study of sRNA in an important human pathogen.

**Part II – Major Issues: Key Experiments Required for Acceptance**

Reviewer #1: Major points:

1) Fig. 1: It would be useful to employ the complete set of wild-type, deletion and complementation strains in all experiments to exclude potential pleiotropic effects. As in Figure 1A, the complementation with CcnA, B and D is fully sufficient to restore wild-type like growth; this is missing for the TIGR4 serotype (Figs. 1D-F).

2) Introduction and the first section of the results part: the authors should highlight the difference between the two serotypes and the reason for using both strains. Why do you see differences between the ∆ccn mutant and the wild-type in the TIGR4 serotype and not in the D39 serotype? Here, it would be useful to compare the results with a complemented strain.

3) Results: please provide a statement on the degree of redundancy among the five Ccn sRNA? Are they different in sequence and structure? And are all of them regulated in the same way by the same two-component system?

4) The time frame of the growth kinetics of S. pneumoniae in various media should be extended (e.g. in Fig 1B). The ∆ccn strain displays a growth defect, but it remains unclear whether the mutant will reach the wild-type OD. Similarly, in Fig 1C it might be possible that the difference between wild-type and the ∆ccn mutant more prominent at later growth stages (see also Figs. 6 and 7).

5) Table 3: The data suggests that the Ccn sRNAs downregulate CzcD expression independently of zinc. Does the abundance of the Ccn sRNAs change after stimulation with high zinc concentrations? This could be tested by northern blotting? How can one explain higher intracellular zinc levels in the ∆ccn mutant, although czcD is highly expressed?

6) Discussion: it would useful to clarify under which conditions metal intoxication with zinc might be relevant for S. pneumoniae (e.g. phagocytosis by macrophages, combination with oxidative stress). Does this play a role for your pneumonia infection model? This link is missing.

Reviewer #2: - The strains shown in Figure 1 include complementation of ccnCD and ccnABD. Have the authors included complementation strains of ccnE? This would be interesting as deletion of ccnE alone resulted in growth deficiency shown in Fig S1.

- The authors note that bacterial burden in the blood of some mice infected with the ccn mutant strain were as low as ~13 CFU/mL. It would be helpful to see any data showing tissue or blood burden for the systemic models of infection. If mice infected with parental and ccn mutant strain bacteria have equal bacterial burden in blood or tissues but are not experiencing symptoms of infection or succumbing to infection, this could be an interesting finding.

- As written, it is confusing to show data prior to discussion of it in the manuscript. For example, all of Fig 1, S1, and S3 are shown with and without the addition of zinc but not fully discussed until Ln 185-196. This results in a feeling of redundancy when you get to the portion of the results section when this is actually discussed. There is similar confusion in the current placement of the sodA Northern blot that is not mention in the results section at Ln 156-168. The easiest solution for this, I think would be moving Fig 2 and S2 ahead of all growth data. The sodA Northern data should be moved to later in the manuscript where Mn-related genes are mentioned and therefore, relevance of sodA is more apparent.

- The statement on Ln 222-224 is an important precursor for the result statement on Ln 165-168. Without this introductory prelude, it’s unclear how Mn fits into this mechanism.

Reviewer #3: none noted.

**Part III – Minor Issues: Editorial and Data Presentation Modifications**

Reviewer #1: Minor points:

Line 28: zinc intoxication and mis-metallation is specific to S. pneumoniae? I recommend to re-write the first two sentences more generally.

Line 115: Why in a rpsL1 background strain?

Line 191: “or” instead of “and”

Line 270-278: redundant

326: swap have and we

Line 333: in “the” Zn

Line 353: strains

Reviewer #2: - Table 2 could be moved to supplement.

- Figure 3 would benefit from statistical analyses.

- Ln 389 – edit ‘invasive pneumoniae’ to ‘invasive pneumonia’

Reviewer #3: (1) when first describing the complement strategy, spell out the rationale for the strategy of generating Dccn//ccnA+B+D+ and Dccn//ccnC+D+ , and why the followed up with the former.

(2) In the description of Figure 5 and Table 2, could the authors address the observation that the complemented strain does not rescue WT phenotype in C+Zn.

(3) Tables with gene expression results (Table S3 & s4) have one column with either name or ID. These are more user friendly, especially for parsing by programs, when one column uniformly has the IDs and another has annotation when available.

PLOS authors have the option to publish the peer review history of their article (what does this mean?). If published, this will include your full peer review and any attached files.

Reviewer #1: No

Reviewer #2: No

Reviewer #3: No
---

## [Decision Letter · Decision Letter 1]

22 Sep 2024

Dear Dr. De Lay,

We are pleased to inform you that your manuscript 'The five homologous CiaR-controlled Ccn sRNAs of Streptococcus pneumoniae modulate Zn-resistance.' has been provisionally accepted for publication in PLOS Pathogens.

Best regards,

Rachel M McLoughlin, PhD

Academic Editor

PLOS Pathogens

Helena Boshoff

Section Editor

PLOS Pathogens

Michael Malim

Editor-in-Chief

PLOS Pathogens

orcid.org/0000-0002-7699-2064

Reviewer Comments (if any, and for reference):

Reviewer's Responses to Questions

**Part I - Summary**

Reviewer #1: The authors addressed all of my previous comments. I think the manuscript is now ready for publication.

Reviewer #2: This is an exciting and innovative study that couples sRNA regulation with metal ion homeostasis, and understudied and likely, emerging field.

Reviewer #3: My comments have all been addressed in this re-submission.

**Part II – Major Issues: Key Experiments Required for Acceptance**

Reviewer #1: (No Response)

Reviewer #2: (No Response)

Reviewer #3: None noted.

**Part III – Minor Issues: Editorial and Data Presentation Modifications**

Reviewer #1: (No Response)

Reviewer #2: (No Response)

Reviewer #3: None noted.

PLOS authors have the option to publish the peer review history of their article (what does this mean?). If published, this will include your full peer review and any attached files.

Reviewer #1: No

Reviewer #2: No

Reviewer #3: No

---

## [Editor Report · Acceptance letter]

27 Sep 2024

Dear Dr. De Lay,

We are delighted to inform you that your manuscript, "The five homologous CiaR-controlled Ccn sRNAs of Streptococcus pneumoniae modulate Zn-resistance.," has been formally accepted for publication in PLOS Pathogens.

Best regards,

Michael Malim

Editor-in-Chief

PLOS Pathogens

orcid.org/0000-0002-7699-2064